# ves1α genes expression is the major determinant of *Babesia bovis*-infected erythrocytes cytoadhesion to endothelial cells

Hassan Hakimi [ID][1,2]*, Junya Yamagishi[3,4], Miako Sakaguchi[5], Atefeh Fathi[1],
Jae Seung Lee[1], Guilherme G. Verocai[2], Shin-ichiro Kawazu[1], Masahito Asada [ID][1]*

**1** National Research Center for Protozoan Diseases, Obihiro University of Agriculture and Veterinary Medicine, Obihiro, Hokkaido, Japan, **2** Department of Veterinary Pathobiology, College of Veterinary Medicine and Biomedical Sciences, Texas A&M University, College Station, Texas, United States of America, **3** Research Center for Zoonosis Control, Hokkaido University, Sapporo, Japan, **4** Global Station for Zoonosis Control, GI-CoRE, Hokkaido University, Sapporo, Japan, **5** Central Laboratory, Institute of Tropical Medicine (NEKKEN), Nagasaki University, Nagasaki, Japan

* hhakimi@cvm.tamu.edu (HH); masada@obihiro.ac.jp (MA)

## Abstract

*Babesia bovis* causes the most pathogenic form of babesiosis in cattle, resulting in high mortality in naive adults. This parasite invades red blood cells (RBCs) within the bovine hosts where they multiply and produce clinical disease. *Babesia bovis* exports numerous proteins into invaded RBCs changing its properties. Thus, the infected RBCs (iRBCs) are capable to cytoadhere in the microvasculature of internal organs and brain, leading to respiratory distress, neurologic signs, and mortality. Variant Erythrocyte Surface Antigen 1 (VESA1) is one of those exported proteins by *B. bovis* which represents a major virulence factor due to its central role in immune evasion by antigenic variation and intravascular parasite sequestration. VESA1 is a heterodimer protein encoded by *ves1α* and *ves1β* multigene family and localized on the ridges, the focal point for cytoadhesion. To gain further insights into the molecular mechanisms of cytoadhesion of *B. bovis*, we panned the parasites with bovine brain microvasculature endothelial cells, which resulted in obtaining several clones with different cytoadherence abilities. The transcriptome analysis of 2 high and 2 low cytoadherent clones revealed that *ves1α* sequences were diversified, likely resulting from genomic recombination. On the other hand, *ves1β* sequences were almost identical among these 4 clones. Insertion and expression of *ves1α* of a clone with high binding into *ef-1α* locus of a low binding clone increased cytoadherence confirming the role of *ves1α* suggested by our transcriptome data. Whole genome sequencing of cytoadherent clones revealed active locus of *ves1* on chromosome 2. These results suggest that VESA1a proteins encoded by *ves1α* genes determine the cytoadherence strength of *B. bovis* and they are in the active site for recombination.

**Data availability statement:** The authors confirm that all data underlying the findings are fully available without restriction. All relevant data are within the paper and its Supporting Information files.

**Funding:** This study was supported partly by a grant from Japan Society for the Promotion of Science (https://www.jsps.go.jp/english/) to H.H. (15K18783, 19K15983), S.K. (18K19258, 19H03120), and M.A. (16K08021, 19K06384). The funders had no role in study design, data collection and analysis, decision to publish, or preparation of the manuscript.

**Competing interests:** The authors have declared that no competing interests exist.

## Author summary

*Babesia bovis* is an apicomplexan intraerythrocytic protozoan parasite that causes the most pathogenic form of babesiosis in cattle. This pathogenicity is the result of parasite multiplication and cytoadherence of infected red blood cells (iRBCs) in the microvasculature of brain and internal organs and is mediated by *B. bovis* surface exposed ligand, Variant Erythrocyte Surface Antigen 1 (VESA1). Here using parasite panning assay, transcriptomics, and genetic tools, we showed that VESA1a is the main determinant of *B. bovis* cytoadhesion. VESA1 are large hypervariable proteins (>100 kDa) consisting of VESA1a and VESA1b subunits encoded by *ves1α* and *ves1β* multigene family. Panning *B. bovis* with bovine brain endothelial cells resulted in obtaining cytoadherent parasite clones with different binding abilities. Comparative transcriptome analysis revealed diversification of *ves1α* sequences. Insertion and expression of *ves1α* of a clone with high-binding ability in the genome of a low-binding clone increased cytoadherence confirming the role of *ves1α*. Mapping RNA-seq on the genome of cytoadherent clones revealed the locus of active transcription and this locus was suggested to be the active site for recombination which promoted the production of variants of *ves1α* with different binding abilities. Altogether, our results provide new insights into *B. bovis* cytoadhesion and VESA1 biology.

## Introduction

Bovine babesiosis is a tick-borne disease with a significant economic impact on the cattle industry. *Babesia bovis*, *B. bigemina*, *B. divergens*, *B. major*, and *B ovata* are the major species responsible for bovine babesiosis and its economic losses world-wide [1–3]. The parasites are transmitted to cattle during the tick blood meal when sporozoites are released from the tick salivary glands and directly invade red blood cells (RBCs). Single forms are divided by binary fission and produce binary forms which subsequently egress and invade new RBCs [4,5]. The cyclical replication of parasites in the erythrocytes causes clinical signs including fever, hemolytic anemia, hemoglobinuria, and neurological syndromes in the case of *B. bovis* which occasion-ally results in fatality [2,6].

Bovine RBCs become rigid following infection with *B. bovis* as the parasite changes the properties of infected RBCs (iRBCs) and reduces their deformability which results in the sequestration of iRBCs [7]. Sequestration of iRBCs with *B. bovis* in the microcirculation of the lung and brain causes cessation of blood flow which produces respiratory distress and cerebral babesiosis that could be fatal if untreated [2,8–10]. Sequestration is mediated by the interaction of iRBC surface protrusions, ridges with bovine microvasculature endothelial cells [8,11]. Sequestration of iRBCs helps the parasites to be protected from oxidative damage in the hypoxic environ-ment of internal organs, avoid spleen clearance and produce a stable infection [12–15].

*B. bovis* is capable of producing asymptomatic, chronic, and persistent infection in cattle which is the outcome of a balance between host immune response and parasite immune evasion [14]. Variant Erythrocyte Surface Antigen 1 (VESA1), the product of the largest multigene family in *B. bovis*, is responsible for antigenic variation and subsequently immune evasion [16,17]. VESA1 is a heterodimeric protein consisting of VESA1a and VESA1b and encoded by *ves1α* and *ves1β* genes, respectively [17–19]. The majority of *ves1α* and *ves1β* genes are located head to head throughout the genome, usually exist in pairs, and one pair of *ves1* is being transcribed at the locus of active transcription (LAT) [20]. It was shown that panning of *B. bovis*-iRBCs for cytoadhesion to endothelial cells selected different isoforms of VESA1 and anti-VESA1a monoclonal antibodies inhibited binding of iRBCs to endothelial cells and reversed binding of cytoadhered iRBCs [11]. VESA1 proteins are clustered on the surface of *B. bovis*-induced iRBC ridges [8,11,21]. The number of ridges increases as the parasites grow and multiply in the RBC [7]. To avoid host antibody-mediated immune response against VESA1 and subsequence clearance of iRBCs, the parasite developed the ability to switch the expression of a different variant of VESA1 likely through epigenetic *in situ* switching or genomic recombination at LAT which results in mosaicism of *ves1* and antigenic variation [14,20].

Although it was shown that VESA1 is involved in cytoadhesion to endothelial cells [11], the functional contribution of VESA1a and VESA1b in cytoadhesion, the binding epitopes of VESA1 to endothelial cells, the receptor of VESA1 on endothelial cells, and other possible players remain to be determined. In this study, we selected the iRBCs for cytoadhesion to bovine brain endothelial cells (BBECs). Several parasite clones were produced with different cytoadhesion abilities. RNA-seq analysis of 2 high-binding and 2 low-binding parasite clones revealed the nucleotide diversity in the expressing *ves1α* gene while the sequences of expressing *ves1β* genes were conserved. Expressing the *ves1α* from a high-binding parasite in the low-binding parasite increased their cytoadherence ability indicating that *ves1α* is likely the main determinant of *B. bovis* cytoadhesion to BBECs. Additionally, *de novo* whole genome sequencing of these clones determined that LAT was located on chromosome 2 and *ves1α* sequence diversity likely happened through recombination which promoted the creation of *ves1* gene chimera with different binding abilities.

## Results

### Selection of *B. bovis*-infected RBCs for adhesion to BBECs is linked with *ves1α* expression

Previously, we selected *B. bovis* Texas strain for cytoadhesion to BBECs [22] by panning an uncloned parasite population. This parasite line has been in the *in vitro* culture for more than 20 years in our institute and did not have the ability to bind to BBECs. Following panning, cytoadherent parasites appeared at the 16th round of selection and we stopped panning at the 21st round when the parasites reached the maximum and stable binding ability. We cloned this cytoadherent parasite line by limiting dilution. Cytoadhesion assay showed that the cloned parasite lines had diverse binding ability from low to high in comparison with parental uncloned cytoadherent parasites (Fig 1A and 1B). To find the genes responsible for cytoadhesion, 4 clones of cytoadherent *B. bovis*, 2 high-binding (C1 and C3) and 2 low-binding (C2 and C5), were subjected to RNA-seq. We considered up- or downregulated genes with more than two-fold change in the expression and a significant *P*-value (P ≤ 0.05). Comparison of transcripts between 2 high and 2 low cytoadherent clones did not reveal any significant hits that have ligand structure or export motif (S1 Table). However, RNA-seq indicated the expression of different variants of VESA1a in 4 clones which likely resulted from genomic recombination or these variants were selected by panning (S1 Fig). The sequences of *ves1α* transcripts from RNA-seq data were further confirmed by primer walking. The expression of a unique *ves1α* was confirmed by qRT-PCR (Fig 2). Interestingly, unlike VESA1a, the VESA1b were almost identical among 4 clones (S2 Fig). Among other candidates that were previously shown to participate in cytoadhesion [22–24], the expression levels of *veap* and *sbp3* were not significantly different between high-binding and low-binding clones (S2 Table). Additionally, *sbp2t11* was not expressed in any of these clones. Altogether, these results suggested that *ves1α* is the main determinant for the cytoadhesion of *B. bovis* to endothelial cells under static conditions.

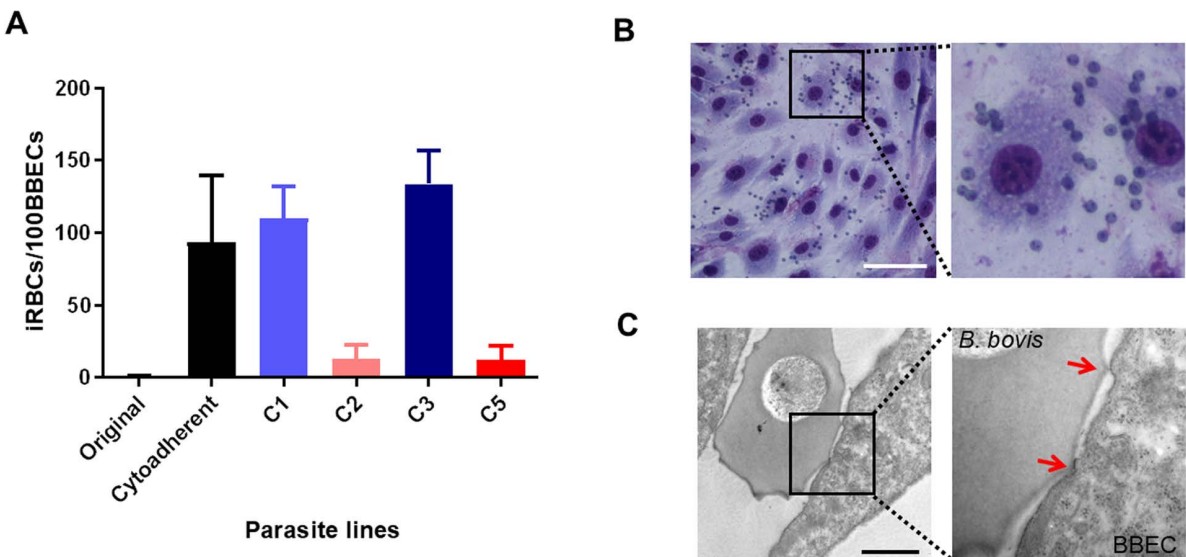

**Fig 1. Panning of *B. bovis* for binding to BBECs. A)** Selection of *B. bovis* for binding to BBECs. The number of bound iRBCs per 100 BBECs was counted. All data are expressed as mean ± SD. The assay was done in duplicate in 3 biological replicates. **B)** Giemsa-stained smear of cytoadherent *B. bovis* to BBECs. Small cells are iRBCs having single or paired forms of *B. bovis*. The larger cells are BBECs. Scale bar = 100 μm. **C)** Transmission electron micrograph of *B. bovis*-iRBC bound to BBEC through ridges (indicated by red arrows). Scale bar = 1 μm.

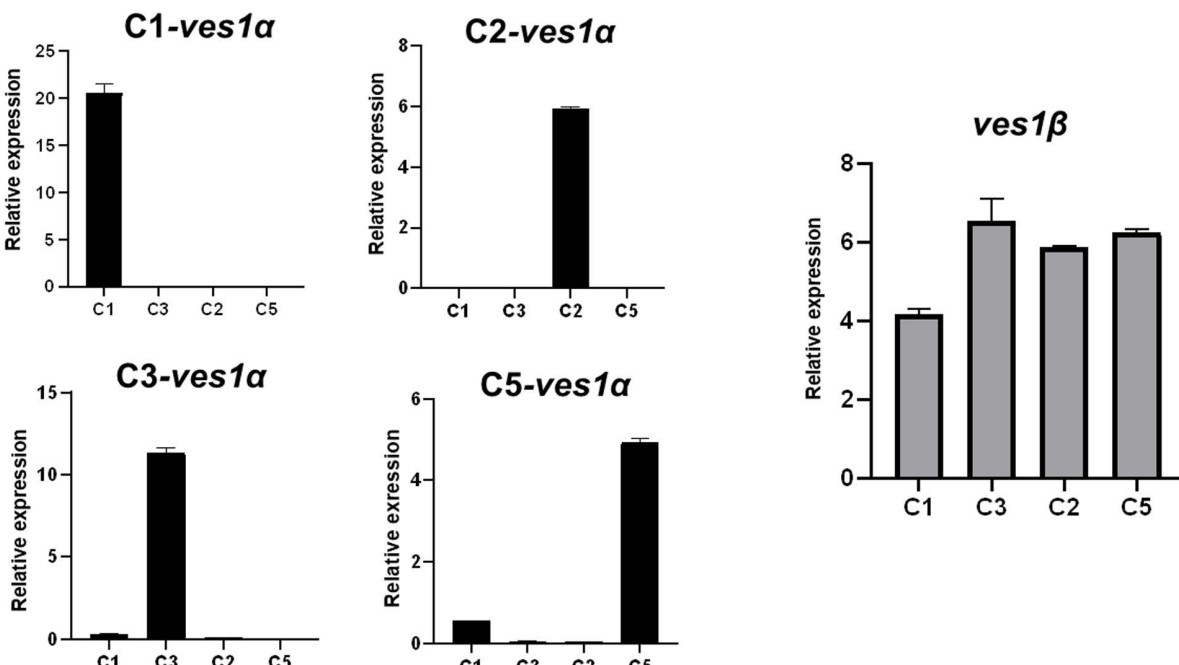

**Fig 2. Quantitative reverse transcription PCR (qRT-PCR) of *ves1α* and *ves1β*.** Absolute transcript levels of *ves1α* and *ves1β* in each cytoadherent parasite clone were quantified using qRT-PCR. All data are shown as mean ± SD. Transcript levels were normalized against *methionyl-tRNA synthetase* (Gene ID: BBOV_I001970).

## VESA1a is the major determinant for binding of iRBCs to BBECs

To confirm *ves1α* as the determinant for cytoadhesion, we performed a gain-of-function study by inserting a *ves1α*-expressing plasmid into the genome. *B. bovis* has two copies of *ef-1α* that are located head-to-head on chromosome 4 [25]. Given that the deletion of one of the two *ef-1α* copies does not affect parasite growth *in vitro* [25], we replaced one copy of *ef-1α* in C2 clone (low-binding clone) with *ves1α-gfp-hdhfr* plasmid expressing *ves1α* from C1, C3 (high-binding clones) or C5 (low-binding clone) (Fig 3A). We were able to get transgenic parasites from transfection with plasmids expressing *ves1α* from C3 or C5. Thus, we characterized these transgenic parasites. Integration of the *ves1α-gfp-hdhfr*-expressing cassette into *ef-1α* locus was confirmed by two diagnostics PCRs (Fig 3A). Western blot analysis confirmed the expression of chimeric VESA1a-GFP protein in the transgenic parasites (Fig 3B). To verify the export and surface expression of VESA1a-GFP, initially, we tried to confirm it using live fluorescence microscopy. However, we were not able to confirm GFP signal at the iRBC membrane which could be due to the low expression of VESA1a-GFP. Therefore, we decided to perform an indirect immunofluorescence antibody test (IFAT) using fixed smears of transgenic parasites episomally overexpressing *ves1α-gfp-hdhfr*. IFAT showed a punctate pattern expression of VESA1a-GFP (S3 Fig) which resembles the expression of VESA1 on ridges. Transgenic parasites expressing

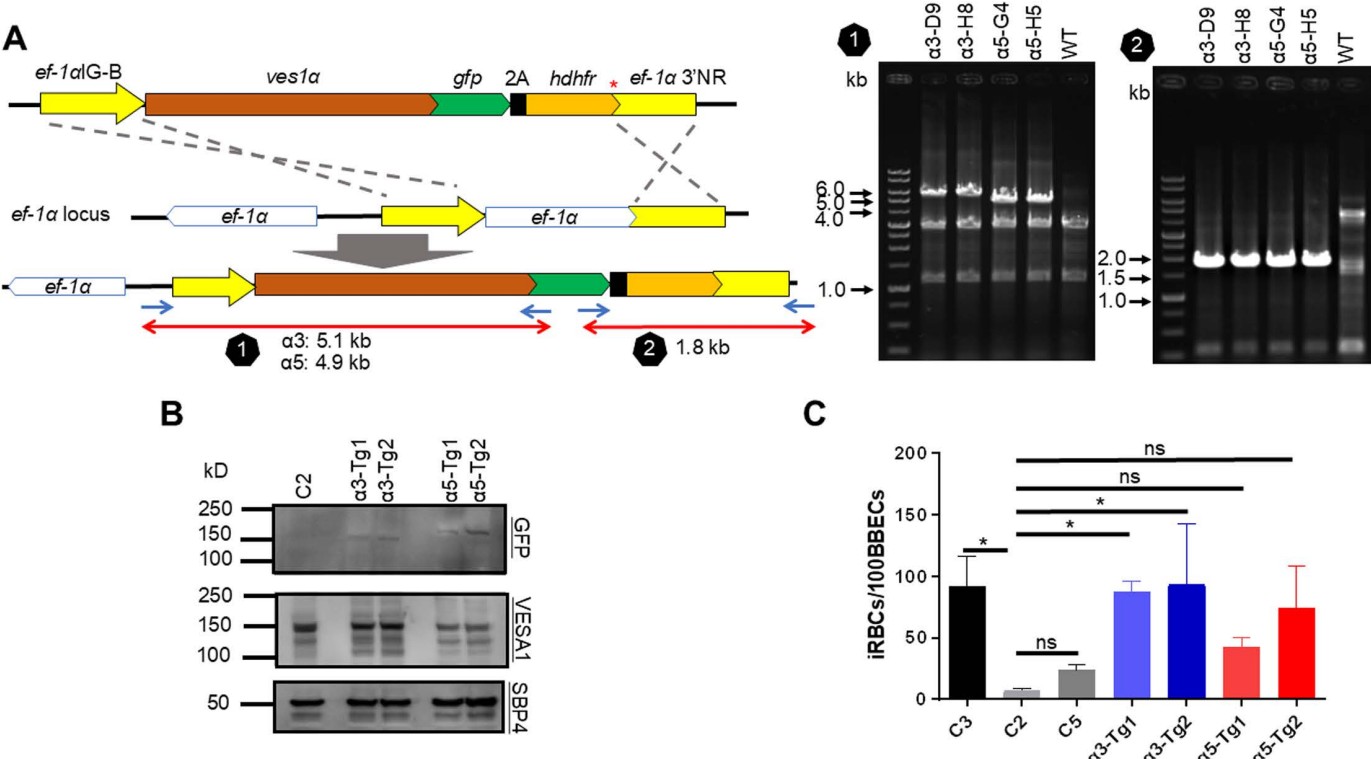

**Fig 3. *ves1α* is the major determinant for binding of iRBCs to BBECs. A)** Schematic diagram of *ves1α-gfp-hdhfr*-expressing plasmid and diagnostic PCRs to confirm integration of *ves1α-gfp-hdhfr*-expressing plasmid into *ef-1α* locus. Stop codon is shown with a red asterisk. *ef-1α*IG-B, *elongation factor-1α* intergenic region B; *ef-1α*-3'NR, *elongation factor-1α* 3' noncoding region; *hdhfr*, *human dihydrofolate reductase*. **B)** Western blot analysis of C2 clone and 2 transgenic parasite clones expressing C3 (α3) or C5 *ves1α* (α5) on C2 clone parasite. Anti-GFP shows the chimeric VESA1-a-GFP expression in the transgenic parasites. Anti-VESA1 and anti-SBP4 antibodies were used to detect VESA1, and SBP4 protein as a loading control. **C)** Cytoadhesion assay of *ves1α-gfp-hdhfr*-expressing parasite clones and WT parasites. C3, C2, C5: original clones before transfection; α3-Tg1 and Tg2: transfectants expressing C3 *ves1α* on C2 clone parasite; α5-Tg1 and Tg2: transfectants expressing C5 *ves1α* on C2 clone parasite. All data are expressed as mean ± SD of three independent experiments with two technical replicates per sample (*, *P* < 0.05; ns, not significant; determined by Tukey's multiple comparison test).

*ves1α* from C3 showed significantly increased cytoadherence to BBECs compared to C2 ([Fig 3C]). Expression of *ves1α* from C5 into C2 also increased binding, though not significant, which could be due to the higher expression of *ves1α* driven by the strong *ef-1α* promoter that needs to be verified in future studies.

To validate any potential contribution of VESA1b to cytoadhesion, we episomally transfected C2 with *ves1β-mCherry-bsd*-expressing plasmids that express *ves1β* from C3 (high-binding clone) or C5 (low-binding clone) ([Fig 4A]). Western blot analysis confirmed the expression of the chimeric VESA1b-mCherry in two transgenic lines generated from each plasmid ([Fig 4B]). We proceeded with these transgenic lines for cytoadhesion assay. No significant increase was seen in binding to BBECs in trans-genic lines compared with C2 confirming that VESA1b had no role in cytoadhesion ([Fig 4C]). Overall, these results suggest that VESA1a is the main player for cytoadhesion.

## Identification of LAT in cytoadherent clones

It is believed that *ves1α* and *ves1β* pair genes are expressed from an LAT in the genome of *B. bovis* and *ves1α* and *ves1β* expression is mutually exclusive from other *ves1α* and *ves1β*, respectively [20]. To find LAT, we performed *de novo* whole genome sequencing of C1, C2, C3, and C5 clones using long-read Oxford Nanopore Technologies sequencing ([Fig 5]). Given that *ves* genes make the largest multigene family with 133 copies in *B. bovis* T2Bo genome (reference strain) [19,26] and have sequence homology, we decided to use long read sequencer to mitigate miss assembly of similar sequences that may happen with short read sequencers. Initially, we performed dot plot analysis between *B. bovis* T2Bo strain and the cytoadherent parasite clones to find the structural variation among the genomes of these parasites ([Fig

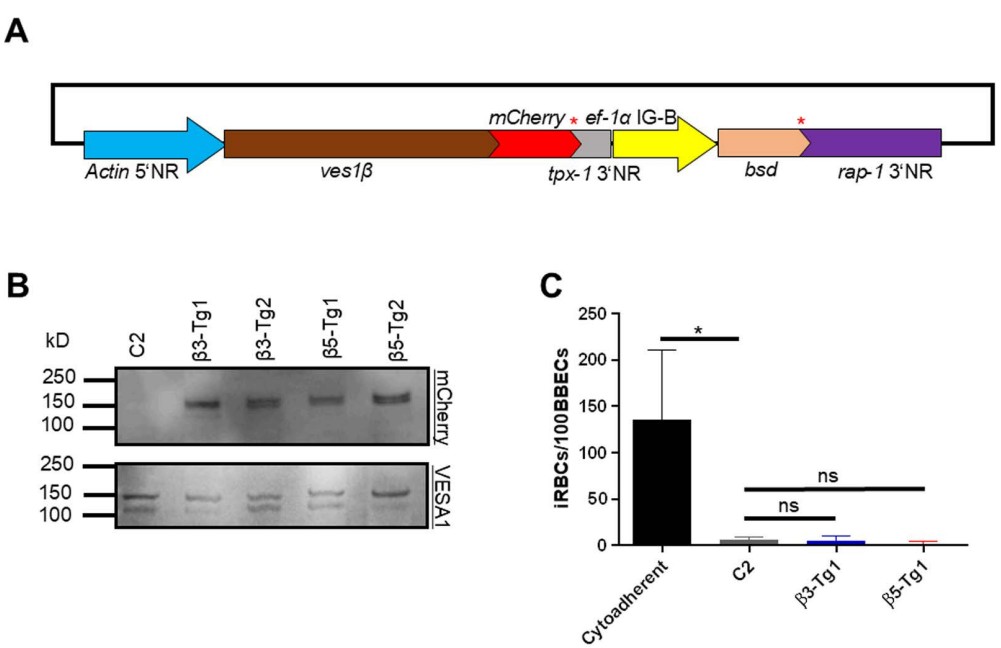

**Fig 4. *ves1β* does not increase binding of iRBCs to BBECs. A)** Schematic diagram of *ves1β-mCherry-bsd*-expressing plasmid. Stop codons are shown with red asterisks. *actin* 5'NR, *actin* 5' noncoding region/promoter; *tpx-1* 3'NR, *thioredoxin peroxidase-1* 3' noncoding region; *ef-1α*IG-B, *elongation factor-1α* intergenic region B; *bsd*, *basticidin S deaminase*; *rap-1* 3' NR; *rhoptry associated protein-1* 3' noncoding region. **B)** Western blot analysis of C2 clone and 2 transgenic parasites episomally expressing C3 (β3) or C5 *ves1β* (β5) on C2 clone parasite. Anti-mCherry shows the chimeric VESA1b-mCherry expression in the transgenic parasites. Anti-VESA1 antibody was used to detect VESA1 protein as a loading control. **C)** Cytoadhesion assay of *ves1β-mCherry-bsd*-expressing parasites and WT uncloned cytoadherent line. β3-Tg1: transfectant episomally expressing C3 *ves1β* on C2 clone parasite; β5-Tg1: transfectant episomally expressing C5 *ves1β* on C2 clone parasite. All data are expressed as mean±SD of three independent experiments (*, *P*<0.05; ns, not significant; determined by Tukey's multiple comparison test).

). The parental parasite of our cytoadherent parasite line was maintained in *in vitro* culture for several years. We found structural variation within chromosomes 1 and 2 between T2Bo and our cytoadherent clones. While clear collinearity of chromosomes 3 and 4 was observed. The synteny map also confirmed recombination events within chromosomes 1 and 2 (Fig 5B). The collinearity among cytoadherent clones was well conserved. Additionally, by mapping acquired reads from C1, C2, C3, and C5 RNA-seq on the corresponding genome, we found that C2 and C3 have a single dominant LAT on chromosome 2 sub-telomeric region that the majority of the RNA-seq reads are mapped on LAT (Figs 5C and 6). However, C1 had 2 LATs, one on chromosome 2 and one on chromosome 3 suggesting the existence of two parasite populations or simultaneous expression of two LATs. We could not find a major LAT for C5 which could be due to miss assembly or lack of clear LAT in this parasite clone. We confirmed the simultaneous expression of *ves1α* and *ves1β* pair genes from LAT which was suggested by the identification and characterization of a bidirectional promoter in the *ves* intergenic region [27]. Further pairwise alignment of *ves1α* transcripts of cytoadherent clones from primer walking and RNA-seq revealed identical results for C3 and matched with the LAT location. However, the pairwise comparison for C1, C2 and C5 gave discordant results. This could be due to the amplification of a *ves1α* transcript from a subpopulation or generation of a *ves1α* chimera during PCR amplification in the primers walking or an assembly error of RNA-seq. Additionally, differences in timing to perform RNA-seq, primer walking, and whole genome sequencing might contribute to these discordant results.

## Discussion

Sequestration of iRBC is a unique feature of *B. bovis* and is principal for its pathogenicity which results in cerebral babesiosis and multiorgan failure [8,14]. The detailed molecular mechanism responsible for *B. bovis* sequestration is not well known. To gain insights into the cytoadherence of *B. bovis*-iRBCs, we performed a panning assay under the static

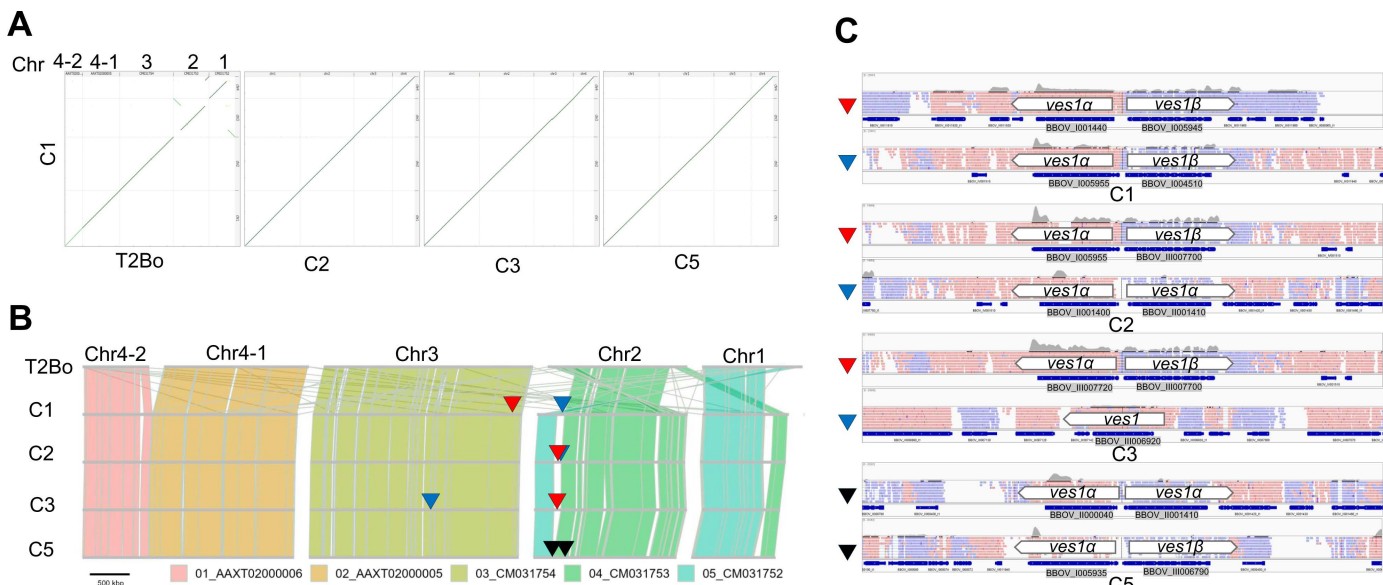

**Fig 5. Genome synteny between *B. bovis* T2Bo and cytoadherent clones. A)** The dot plot analyses among reference T2Bo, C1, C2, C3, and C5 identified genomic recombination. Dot plots were generated by D-Genies. The diagonal line shows the alignment between two parasite genomes. Short off-diagonal lines between C1 and T2Bo show structural variations in chromosomes 1 and 2. **B)** The synteny blocks of T2Bo were aligned with those of cytoadherent clones. Chromosome 4 of T2Bo has a gap and consists of two contigs (Ch4-1 and Ch4-2). The primary and alternative LATs are indicated with red and blue triangles, respectively. **C)** Integrative genomics viewer (IGV) of candidate LATs in cytoadherent clones. LATs were identified by aligning the RNA-seq reads on the genome of corresponding cytoadherent clones sequencing using MinION. The primary and alternative LATs are indicated with red and blue triangles, respectively. The aligned *ves1* reads (black triangles) on the C5 genome did not reveal a major LAT.

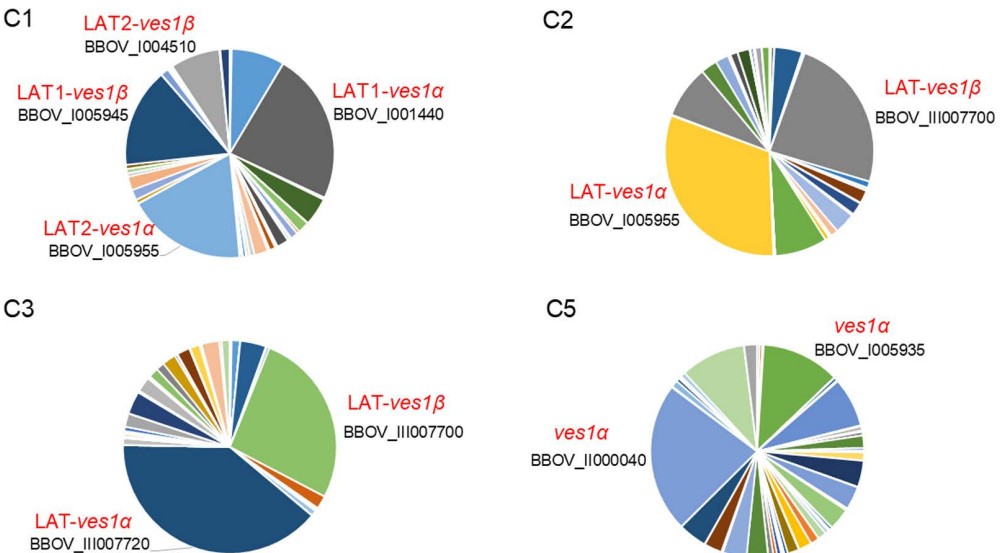

**Fig 6. Pie chart of expressing *ves* genes in the cytoadherent clones.** Pie charts display *ves* genes expression profile of each cytoadherent clone population with each slice of the pie representing the expression level of a single *ves* gene. The expressing dominant *ves1α* and *ves1β* from a single LAT are labeled in each clone. Gene IDs are from T2Bo reference strain and are assigned based on *ves1* sequence similarity.

condition as described [21]. We were able to select cytoadherent parasite clones with high and low binding abilities. Comparative transcriptomics among 2 high and 2 low-binding parasite clones revealed that *ves1α* sequences were diversified while *ves1β* sequences were conserved suggesting *ves1α* is the main determinant of cytoadhesion. These findings were supported by a gain-of-function study where we inserted *ves1α* orf from a high (C3) or low binding clone (C5) into the *ef-1α* locus of C2 clone as a low binding and the background parasite for transfection. Transgenic parasites expressing *ves1α* orf from C3 showed significantly higher binding to BBECs compared to parental parasite C2 confirming that *ves1α* is responsible for cytoadherence. Although it was shown that a specific monoclonal antibody against VESA1a is able to inhibit cytoadhesion and reverse the binding of cytoadherent iRBCs [11], here for the first time using genetic tools we showed that VESA1a is the main ligand for cytoadhesion. Given that *ves1α* in transgenic lines is driven by the strong promoter of *ef-1α*, VESA1a protein may have been expressed at times and/or levels that are not reflective of the natural situation. VESA1 is a heterodimeric protein consisting of VESA1a and VESA1b encoded by *ves1α* and *ves1β* genes, respectively [17,18]. The molecular structure of VESA1, the interaction between VESA1a and VESA1b, and their assembly on the ridges of iRBCs are not known. Our results suggest that VESA1a is responsible for the binding of VESA1 to the receptor on the BBEC while VESA1b might have a structural function in this complex. The molecular structure of ridges and the proteins responsible for ridge formation is unknown. Given that VESA1b similar to VESA1a is an integral protein and has a transmembrane domain [18], it is likely that VESA1b participated in ridge formation and their assembly on the surface of iRBCs promotes ridge formation or assist the assembly of VESA1a on the surface of iRBC. Both VESA1a and VESA1b have a cysteine and lysine-rich domain (CKRD) at the N-terminus, however, VESA1a has variant domain conserved sequences (VDCS) in the middle of protein [14]. Finding the binding epitope on VESA1a will further shed light on the functional divergence of VESA1a and VESA1b. Future studies should focus on the identification of binding receptor on BBECs and the application of cryoEM to shed light on the VESA1a-receptor complex structure. We cannot rule out the possibility that VESA1a expression level may contribute to cytoadhesion ability as all transgenic parasite lines express C2 type VESA1a (S4 Fig). A better genetic tool for controlled expression of *ves1α* is needed to test this hypothesis.

Cerebral babesiosis caused by *B. bovis* is similar to cerebral malaria caused by *Plasmodium falciparum*, the most pathogenic agent of human malaria [12]. PfEMP1 is the functional ortholog of VESA1 in *P. falciparum*, responsible for

antigenic variation and cytoadherence. Cellular adhesion of *P. falciparum*-iRBCs consists of a rolling interaction with ICAM1 and a stationary interaction with CD36 and EPCR on vascular endothelial cells which leads to severe malaria [28–30]. It was shown that under flow conditions, *B. bovis*-iRBCs can make strong and stable interactions [7]. Although in this study, using a static cytoadhesion assay, we were not able to find novel ligands for binding of iRBCs, future application of assay under flow condition which is physiologically more relevant may lead to finding novel ligands [7]. Recently, two secreted proteins were shown to contribute to cytoadhesion, SBP2 truncated copy 11 and VESA1-Export Associate Protein, BbVEAP [22,24]. SBP2 truncated copy 11 was shown to be upregulated at transcriptome and protein levels in an attenuated *B. bovis* strain and its overexpression resulted in decreased binding of iRBCs to endothelial cells [24,31]. However, the mechanism behind this finding is unclear. BbVEAP expression was shown to be essential for VESA1 export and expression on iRBCs, and normal formation of ridges, however, no direct interaction was found between VESA1 and BbVEAP [22]. Recent developments in molecular genetic tools for *B. bovis* [25,32,33] enable us to verify the possible role of VESA1 in cytoadhesion and gain more insights into this interaction.

To find the position of LATs in cytoadherent clones, we performed whole genome sequencing using long-reader MinION and aligned the RNA-seq reads on the corresponding parasite clone genome. Finding LAT location will facilitate understanding antigenic variation of *B. bovis* and will be a foundation for developing novel control strategies targeting antigenic variation. We found that C2 and C3 express a major pair of *ves1α* and *ves1β* located in LAT. Our findings are aligned with a previous study that used universal primers to check the expression of *ves1α* genes and discovered mutual exclusive transcription of these genes [34]. Additionally, we found that LAT exist on chromosome 2 for several of these cytoadherent clones. Further analysis of genome structure among cytoadherent clones and the reference T2Bo strain confirmed that LAT locus is the active site for recombination which promotes the recombination of different *ves1α* genes and ultimately results in the production of *ves1α* chimera. Genomic recombination at the LAT locus happens through segmental gene conversion and produces mosaicism of *ves1α* which ultimately assists the parasite in evading immune response and establishing a persistent infection [14,20,35]. Although we found variant *ves1α* transcripts, the sequences of expressing *ves1β* from LAT were conserved. This finding was surprising as both expressing *ves1α* and *ves1β* genes are in the same location on chromosome. The *ves1* multigene family of *B. bovis* has 81 *ves1α*, 48 *ves1β*, and 4 unclassified *ves1* [19,26]. Whether a specific motif or sequence within *ves1α* is promoting recombination needs further investigation. The frequent recombination characteristic of this multigene family is likely limited to *ves1α* which is explained by the function of VESA1a for cytoadhesion.

The *ves* multigene family is conserved across *Babesia* sensu stricto, though their sequence is diversified. *B. bigemina* and *B. ovata* express *ves1a*, *ves1b*, and *ves2* which share sequence homology between these two parasites [13,36,37], while *B. divergens* and *Babesia* sp. Xingjang have their unique *ves* in their genome [37,38]. Recently, we reported the expansion of *ves1c* genes in *B. caballi* and showed that their expression is controlled through epigenetic monoallelic expression as the majority of *ves1c* transcripts mapped to a single gene [39]. Although small ruminants infecting *B. ovis* is phylogenetically close to *B. bovis*, its *ves* genes sequences are diversified from the ones of *B. bovis* [40]. Due to simple structure of *ves1α* (having 2 introns) in comparison with *ves1β* (having up to 11 introns), it was proposed that *ves1α* may have originated from incompletely spliced *ves1β* transcript through retrotranscription [17,20]. Given that the unique biology of *B. bovis* sequestration is linked to *ves1α*, it is plausible to assume that *B. bovis* *ves1β* kept the evolutionary conserved function of *ves* genes across the *Babesia* sensu stricto. Overall, the findings of this study will contribute to our understanding of *B. bovis* *ves1* genes biology, cytoadhesion, and immune evasion.

## Methods

### *B. bovis in vitro* culture

*B. bovis* Texas strain [41] was maintained *in vitro* using a microaerophilic stationary-phase culture system composed of purified bovine RBCs (Japan BioSerum, Fukuyama, Japan) at 10% hematocrit and GIT medium (Wako Pure Chemical Industries, Japan). Parasitemia was monitored by preparing thin smears that were fixed with methanol and stained with

Giemsa's solution. *In vitro* cultured parasites were cryopreserved in 20% polyvinylpyrrolidone 40K (w/v) (P0930, Sigma-Aldrich) in the Vega and Martinez solution as described [42].

## Plasmid construction

The schematic of the *ves1α-gfp-hdhfr*-expressing plasmid is shown in Fig 3A. The primers used for the construction of the plasmid are listed in S3 Table. *B. bovis elongation factor-1α* intergenic region-B (*ef-1α* IG-B) and *ef-1α* 3' noncoding region (NR) were PCR amplified from *B. bovis* genomic DNA. *ves1α* full length of coding sequence from C1, C3, and C5 were PCR amplified from respective cDNA. *Human dihydrofolate reductase* (*hdhfr*) and *gfp* were amplified from a *B. bovis* green fluorescent protein (GFP)-expressing plasmid [32]. *hdhfr* and *ef-1α* 3'NR were cloned into the BamHI site of pBluescript SK using In-Fusion HD Cloning Kit (Takara Bio Inc., Otsu, Japan). Subsequently, *ef-1α* IG-B was cloned into the HindIII site. Finally, *ves1α* and *gfp* were cloned into the SmaI site to make the final plasmid. The plasmid was linearized by ApaI and NotI digestion to promote integration into *ef-1α* locus.

The schematic of the *ves1β-mCherry-bsd*-expressing plasmid is shown in Fig 4A. *B. bovis actin* 5' NR/promoter, *thioredoxin peroxidase-1* (*tpx-1*) 3'NR, *ef-1α* IG-B and *rhoptry associated protein-1* (*rap-1*) 3' NR were PCR amplified from *B. bovis* genomic DNA. The full length coding sequence of *ves1β* from C3 and C5 were PCR amplified from respective cDNA. Blasticidin S deaminase (bsd) was amplified from pBrfp-bsd plasmid [43] and *mCherry* was amplified from a *mCherry*-expressing plasmid [44]. The *actin* 5'NR, *ves1β*, *mCherry* and *tpx-1* 3'NR were cloned into the XhoI site of pBluescript SK. Subsequently, *ef-1α* IG-B, *bsd* and *rap-1* 3' NR were cloned into the SalI site. The final plasmid constructs were sequenced by Sanger sequencing to confirm that the cloned elements are in frame and the cloned *ves1α* and *ves1β* are from the respective cytoadherent clones.

Diagnostic PCR was done to confirm the insertion of *ves1α-gfp-hdhfr* into *ef-1α* locus. Two sets of primers were used to amplify the DNA fragments surrounding the 5' recombination site and the 3' recombination site. To examine the 5' recombination event of the plasmid construct, a primer pair Bbef-1a-seqF1 and GFP-seqR was used to amplify 5.1 and 4.9 kb DNA fragments for transgenic parasites expressing *ves1α* from C3 and C5, respectively. To confirm the 3' recombination event of the plasmid construct, a primer pair GFP-seqF and Bbef1-a2KO-3GI-R was used to amplify a 1.8 kb DNA fragment.

## *B. bovis* transfection

*B. bovis* transfection was done as described [32,43]. Briefly, *B. bovis*-iRBCs were washed twice with PBS and once with cytomix buffer (120 mM KCl, 0.15 mM CaCl$_2$, 10 mM K$_2$HPO$_4$, 10 mM KH$_2$PO$_4$, 25 mM HEPES, 2 mM EGTA, 5 mM MgCl$_2$, 100 g/mL bovine serum albumin, 1 mM hypoxanthine; pH 7.6). One hundred μL of *B. bovis*-iRBCs were mixed with 10 μg of plasmid constructs (dissolved in 10 μL of cytomix buffer) and then mixed with 90 μL of Amaxa Nucleofector human T-cell solution. Transfection was done using a Nucleofector device, program v-024 (Amaxa Biosystems, Germany). WR99210 (10 nM) was added 1 day after the transfection *ves1α-gfp-hdhfr*-expressing plasmid to the culture to select a transgenic parasite population. For the selection of transgenic parasites episomally expressing *ves1β-mCherry-bsd*-expressing plasmid, 4 μg/mL of Blasticidin-S (Invitrogen) was added to the culture medium 1 day after the transfection.

## Selection of *B. bovis*-iRBCs for binding to BBEC and cytoadhesion assay

Panning the parasites for binding to bovine brain endothelial cells (BBECs; Cell Applications Inc., USA) and cytoadhesion assays were performed as described [21]. Briefly, BBECs were seeded in 6 well plates in the growth medium. Following 2–3 days of culture when the cells became confluent, *B. bovis*-iRBCs with 2–5% parasitemia and 1% hematocrit were added to the BBEC culture. The cells were incubated for 90 min with gentle agitation every 15 min. Nonadherent iRBCs were removed by washing with Hanks-balanced salt solution. Fresh uninfected RBCs were added to each well and incubated overnight. The RBC suspension was removed and transferred to a new plate and cultured under normal condition.

Once parasitemia reached 2–5%, a new round of selection was performed. The cytoadherent parasite line from the 21st round of panning was cloned by limiting dilution.

For cytoadhesion assays, BBECs were seeded in 6 well plates containing cover glasses (Matsunami Glass, Japan). Once cells become confluent, iRBCs were added and incubated as described. Noncytoadherent RBCs were washed and cells on the cover glasses were fixed with methanol for 5 min and stained with Giemsa's solution. Bound iRBCs were counted for 500 BBECs. Cytoadhesion assays were done three times with two technical replicates per sample. The parasites were maintained in the culture for no more than 5 passages to minimize the chance of *ves1* recombination and switching.

### DNA extraction and whole genome sequencing

Genomic DNA was extracted from C1, C2, C3 and C5 *in vitro* cultured clones with parasitemia of 5–10% using Nucle-oSpin Blood QuickPure kit (Macherey-Nagel GmbH, Düren, Germany). The library for MinION was constructed with a Ligation Sequencing Kit, SQK-NBD114–96 (Oxford Nanopore Technologies), and then sequenced using FLO-PRO114M flow cells (Oxford Nanopore Technologies). The reads obtained from MinION were assembled by wtdbg2 [45]. The obtained assemblies were polished by pilon [46]. The dot plot among T2Bo ver-63 and C1, C2, C3, and C5 was made by D-Genies [47]. The synteny map was made as follows. Synteny blocks identified by MCScanX [48]. The result was visualized by AccuSyn. Annotation of cytoadherent clones was made by liftoff [49] referring to annotation of the T2Bo reference strain.

### RNA extraction and RNA-seq

IRBCs with parasitemia of 5–10% were treated with 0.2% (w/v) saponin on ice for 15 min to remove hemoglobin. Total RNA was extracted from parasite pellets using TRIzol reagent (Thermo Fisher Scientific) following the manufacturer's instructions. Libraries were constructed using a TruSeq Stranded mRNA Library Preparation Kit (Illumina, USA) and the products were subjected to Novaseq6000 (Illumina) with the 150-bp paired-end protocol. Acquired reads were mapped against the obtained genome assemblies using TopHat2 [50]. Relative expression of *ves1α* and *ves1β* was counted by HTseq [51].

### Quantitative reverse transcriptase PCR (qRT-PCR)

To confirm the expression of *ves1α* and *ves1β* in each parasite clone qRT-PCR assays were performed. Following extraction, RNA was treated with DNase I (Promega, Madison, WI, USA) and purified using an SV Total RNA Isolation System (Promega) kit. Complementary DNA (cDNA) was synthesized using SuperScript III Reverse Transcriptase (Invitrogen) using random primers. Parasite clone-specific primers were designed for *ves1α* and one pair of primers for *ves1β* based on RNA-seq and primer walking results (S2 Table). qRT-PCR was performed using Power SYBR™ Green PCR Master Mix (Thermo Fisher Scientific) using a 7500 Real-Time PCR system (Applied Biosystems, Foster City, CA, USA). Transcript levels were normalized against *methionyl-tRNA synthetase* (Gene ID: BBOV_I001970) [22].

### Western blotting

IRBCs were treated with 0.2% saponin to remove hemoglobin and parasite proteins were extracted from parasite pellets using 1.0% Triton-X 100 (w/v) in PBS and protease inhibitor cocktail (Complete Mini, Roche) at 4°C for 1 h. Extracted proteins were separated by electrophoresis on a 5–20% SDS-polyacrylamide gradient mini gel (ATTO, Tokyo, Japan) in a reducing condition, and transferred to polyvinylidene difluoride membranes (Clear Blot Membrane-P, ATTO, Tokyo, Japan). The membrane was probed with rabbit anti-GFP polyclonal antibody (1:500; ab6556, Abcam) or rabbit anti-mCherry polyclonal antibody (1:1000; ab167453, Abcam) or rabbit anti-SBP4 polyclonal antibody (1:1000) [52] or rabbit anti-VESA1a peptide antisera (1:50) [22] at 4 °C overnight. The membranes were incubated with HRP-conjugated goat

anti-rabbit IgG (1:8000; Promega, USA) as the secondary antibody. Bands were visualized using Immobilon Western Chemiluminescent HRP substrate (Merck Millipore) and detected by ImageQuant LAS 500 (GE Healthcare).

### Indirect immunofluorescence antibody test (IFAT)

IFATs were performed to confirm the expression and export of VESA1a-GFP chimeric protein. Thin blood smears from cultured parasites were prepared, air-dried and fixed in a 1:1 acetone:methanol mixture at −20°C for 5 min [53]. Smears were blocked with PBS containing 10% normal goat serum (Invitrogen) at 37 °C for 30 min and immunostained with rabbit anti-GFP polyclonal antibody (ab6556, Abcam) at 1:500 dilution in PBS supplemented with 0.05% Tween-20 and incubated at 4 °C overnight. The smears were incubated with Alexa Fluor 488-conjugated goat anti-rabbit IgG secondary antibody (1:500; Invitrogen) at 37 °C for 30 min. For staining of nuclei, the smears were incubated with 1 μg/mL Hoechst 33342 solution at 37°C for 20 min. Images were obtained using a confocal laser-scanning microscope (CS-SP5, Leica Micro-system, Wetzlar, Germany).

### Transmission electron microscopy

Cytoadhesion assay was done to allow the binding of iRBCs to BBECs. Cells were fixed with 2% glutaraldehyde (Nacalai Tesque, Japan) in 0.1 M sodium cacodylate buffer containing 1 mM $CaCl_2$ and 1 mM $MgCl_2$ at RT for 10 min. Cells were scraped from the culture flask using a cell scraper and moved to 15 mL tubes. The cells were further fixed in the same buffer for an additional 50 min on ice. The cells were post-fixed with 1% $OsO_4$ (Nakalai Tesque) for 60 min on ice, then they were washed, dehydrated in a graded series of ethanol and acetone, and embedded in Quetol 651 epoxy resin (Nisshin EM, Japan). Ultra-thin sections were stained and examined at 80 kV under a transmission electron microscope (JEM-1230; JEOL, Japan).

### Statistical analyses

The number of bound iRBCs to BBECs was plotted using GraphPad Prism 8. The values were considered to be significantly different from the control if $P$-value was below 0.05 using one-way ANOVA followed by Tukey's multiple comparison test.

### Supporting information

**S1 Fig.** Multiple amino acid sequence alignment of VESA1a using Clustal Omega. Comparison of VESA1a sequence revealed by RNA-seq and primer walking among cytoadherent clones. Cysteine and lysine-rich domain (CKRD) and cytoplasmic region are underlined. The transmembrane domain is (TM) boxed.
(TIFF)

**S2 Fig.** Multiple amino acid sequence alignment of VESA1b using Clustal Omega. Comparison of VESA1b sequence revealed by RNA-seq among cytoadherent clones. Cysteine and lysine-rich domain (CKRD) and cytoplasmic region are underlined. The transmembrane domain is (TM) boxed.
(TIFF)

**S3 Fig.** Immunofluorescence microscopy of transgenic parasites episomally expressing VESA1a-GFP. The parasites were reacted with anti-GFP antibody (green) and nuclei were stained with Hoechst 33342 (Hoechst, blue). All the signals were taken at the same focal plane. Scale bar = 5 μm.
(TIFF)

**S4 Fig.** Quantitative reverse transcription PCR (qRT-PCR) of *ves1a*. Relative transcript levels of C2 type *ves1a* in C2, C3, C5, and transgenic parasites were quantified using qRT-PCR. α3-Tg1 and Tg2: transfectants expressing

C3 *ves1α* on C2 clone parasite; α5-Tg1 and Tg2: transfectants expressing C5 *ves1α* on C2 clone parasite. All data are shown as mean ± SD. Transcript levels were normalized against *methionyl-tRNA synthetase* (Gene ID: BBOV_I001970).
(TIFF)

**S1 Table. Genes up- or downregulated in high (C1 and C3) compared to low cytoadherent clones (C2, C5).**
(XLSX)

**S2 Table. Expression of *veap*, *sbp2t11* and *sbp3* in cytoadherent clones.**
(XLSX)

**S3 Table. List of primers used in this study.**
(XLSX)

**S1 Data. Numerical data underlying the graphs shown in this study.**
(XLSX)

## Acknowledgments

This work was conducted at the National Research Center for Protozoan Diseases, Obihiro University of Agriculture and Veterinary Medicine, Hokkaido, Japan. This work was partially conducted at the Joint Usage/Research Center on Tropical Disease, Institute of Tropical Medicine (NEKKEN), Nagasaki University, Nagasaki, Japan.

## Author contributions

**Conceptualization:** Hassan Hakimi, Masahito Asada.

**Formal analysis:** Hassan Hakimi, Junya Yamagishi, Miako Sakaguchi, Atefeh Fathi, Jae Seung Lee, Guilherme G. Verocai, Shin-ichiro Kawazu, Masahito Asada.

**Funding acquisition:** Hassan Hakimi, Shin-ichiro Kawazu, Masahito Asada.

**Investigation:** Hassan Hakimi, Junya Yamagishi, Miako Sakaguchi, Atefeh Fathi, Jae Seung Lee.

**Methodology:** Hassan Hakimi, Junya Yamagishi, Miako Sakaguchi.

**Project administration:** Hassan Hakimi, Masahito Asada.

**Resources:** Hassan Hakimi, Shin-ichiro Kawazu, Masahito Asada.

**Supervision:** Masahito Asada.

**Validation:** Hassan Hakimi, Masahito Asada.

**Visualization:** Hassan Hakimi, Junya Yamagishi, Miako Sakaguchi, Masahito Asada.

**Writing – original draft:** Hassan Hakimi.

**Writing – review & editing:** Hassan Hakimi, Junya Yamagishi, Miako Sakaguchi, Atefeh Fathi, Jae Seung Lee, Guilherme G. Verocai, Shin-ichiro Kawazu, Masahito Asada.

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
