## [Decision Letter · Decision Letter 0]

9 Oct 2024

Dear Dr Hakimi,

Thank you very much for submitting your manuscript "*ves1α* genes expression is the major determinant of *Babesia bovis* -infected erythrocytes cytoadhesion to endothelial cells" for consideration at PLOS Pathogens. As with all papers reviewed by the journal, your manuscript was reviewed by members of the editorial board and by several independent reviewers. In light of the reviews (below this email), we would like to invite the resubmission of a significantly-revised version that takes into account the reviewers' comments.

We cannot make any decision about publication until we have seen the revised manuscript and your response to the reviewers' comments. Your revised manuscript is also likely to be sent to reviewers for further evaluation.

Sincerely,

Tracey J. Lamb

Section Editor

PLOS Pathogens

Tracey Lamb

Section Editor

PLOS Pathogens

Michael Malim

Editor-in-Chief

PLOS Pathogens

orcid.org/0000-0002-7699-2064

Reviewer's Responses to Questions

**Part I - Summary**

Reviewer #1: Babesiosis due to B. bovis shares many similarities with human falciparum malaria, including co-dependence of the antigenic variation and cytoadherence phenomena. Prior work has suggested that cytoadherence in B. bovis depends upon a heterodimeric protein, VESA1, that also mediates rapid antigenic variation. The authors here provide further evidence supporting a link between antigenic variation and cytoadherence mediated by VESA1. Based upon observed sequences in parasites selected for cytoadherence, the authors propose that the VESA1a subunit specifically is responsible for binding. Support for this idea was provided through gain-of-function experiments. The title and abstract accurately represent manuscript content, the figures are of a high quality, the experimental work was done well technically, and the manuscript is overall well-written with correct English usage. However, I have significant reservations about the design and results presented for the gain-of-function experiments underlying the authors’ conclusions regarding the importance of VESA1a, which I feel are ultimately unsupported.

Reviewer #2: This is a report from a leading group in the field of Babesia. They have used the panning technique to generate four Babesia clones with differing affinity to Bovine Brain Endothelial Cells (BBEC) in a static binding assay. Evaluation of the sequence heterogeneity of the two chains of the ves genes of these clones showed that ves1a was highly diversified whereas ves1b showed near identity despite binding affinity differences. They have concluded that the ves1a gene is responsible for binding affinity. Transfection of the ves1a high binding gene into a low binding clone resulted in a statistically significant increase in binding to endothelial cells. A second portion of the manuscript focuses on the identification of the locus of active transcription (LAT) for the ves genes.

The experiments are performed well and presented clearly.

Reviewer #3: The manuscript "Ves1α genes expression is the major determinant of Babesia bovis infected erythrocytes cytoadhesion to endothelial cells" is well-written and presents interesting results. The authors utilized bovine brain endothelial cells to investigate the molecular mechanisms of cytoadhesion of B. bovis. They selected clones with the ability of cytoadhesion using bovine brain endothelial cells. Furthermore, the authors conducted transcriptome analysis of the clones and whole genome sequences.

Please take note of the following concerns regarding the manuscript: This manuscript is based on cloning Babesia parasites to test whether VESA1a is responsible for cytoadhesion. The description of the experiment lacks details important for the conclusion.

1) Cloning a single infected RBC from bovine brain endothelial cells containing multiple attached RBCs is not clearly explained.

2) The authors did not specify the number of passages of the Babesia culture after cloning the parasites.

3) It is unclear why the authors did not select a clone that lacked cytoadhesion ability. This would have allowed them to demonstrate the restoration of cytoadhesion capability by transfected clones.

4) The authors did not provide information on how they restored the RBC culture with the cytoadhesion clones, nor did they specify the parasitemia used for RNA and DNA isolation.

**Part II – Major Issues: Key Experiments Required for Acceptance**

Reviewer #1: Major concerns

1. Gain-of-function experiments. While this is a logical, powerful approach and the centerpiece of the authors’ argument, there are significant flaws in the way this experiment was performed. Some data are also unconvincing. (i) Figure 3B. The results of the western blots are internally inconsistent. Anti-GFP results suggest the presence of VESA1a-GFP fusions of about 160 kDa in a3-Tg1 and 2, and about 180 kDa in a5-Tg1 and 2. In contrast, anti-VESA1 results indicate a single, major VESA1a protein of consistent size (about 150 kDa) in the C2 parental line and each of the selected variant clones. No evidence of the GFP fusion protein is seen despite its presumed overexpression. Please explain. (ii) Figure 3C. It is unclear why the authors would choose to use the cytoadherent C2 line to perform gain-of-function experiments for the adherent phenotype when this line is already cytoadherent. This choice is illogical, particularly when non-adhesive lines could have been obtained for this use, and may underlie the highly varied results obtained with the various transgenic parasite lines (alternatively, both C2 and a non-adhesive line could be engineered for contrast). (iii) A control transgenic line in which a non-adhesive ves1a gene was similarly expressed should have been included to demonstrate a failure to provide this function. Similarly, ves1b genes could have been used to demonstrate (presumed) failure to impart the cytoadherent phenotype. (iv) The authors present no data confirming that the original C2 ves1a gene is being expressed in the transgenic lines, while attributing all binding to the inserted ves1a genes. Given the parasite’s rapid antigenic variation, transgenic lines may or may not still be expressing the original C2 ves1a gene. This seriously compromises interpretation of the results. (v) The C2 ves1a itself should also be inserted into the EF1a site to assess the effects of VESA1a/VESA1b stoichiometry on cytoadherence. Without the above experiments I do not believe it is possible to make rigorous interpretations regarding the contributions of either VESA1a or 1b in the cytoadherence function. (vi) The inserted ves1a genes are likely transcribed at high levels throughout the cell cycle because of the highly active EF1a promoter. VESA1a protein may have been expressed at times and/or levels that are not reflective of the natural situation, requiring caution in interpreting the results.

2. line 150. An alternative and potentially more plausible interpretation is that VESA1b is responsible for the cytoadherence phenotype, since selection resulted in the expression and/or maintenance of essentially the same form of the protein, whereas significant changes in the sequence of VESA1a did not have a qualitative effect. Unfortunately, the authors did not provide pre-selection sequences or sequences from non-cytoadherent lines to know what was already being transcribed or associated with non-adhesive phenotype, nor was this possibility addressed in the gain-of-function experiments. It is interesting to note from Figure 2 that the stoichiometry of ves1a/ves1b transcripts is much higher in the C1 and C3 high-adhesion lines than in the C2 and C5 low-adhesion lines. Stoichiometry of the two subunits may be significant to expression of the adhesive function.

3. Figure 3C. The mapping of reads to the sequences included in the figure is sometimes not convincing. How confident are the authors that they have identified the correct expression site? Some, such as the “blue” C5 ves1a sequences appear as though a short sequence may have come from that site but also be a part of a different, complete, transcribed gene. This is a pattern one might see when some sequences in the transcribed gene are derived by segmental gene conversion. This affects interpretation of LAT identity, monoallelic exclusion, etc.

Reviewer #2: Three major concerns dampen overall enthusiasm.

1) Lines 143 and 144 state that “Comparison of transcripts between 2 high and 2 low cytoadherent clones did not reveal any significant hits (Sup Table 1).”, yet I see eight genes with expression upregulated and nine genes with expression downregulated with increased binding (all with p-values <0.05 – some with p-values as low as 10^-16) in this Table. It is unclear why these genes are not discussed and instead the ves genes are the focus of investigation.

2) Although briefly discussed, the concern of binding affinity to BBEC being merely a matter of expression level (of any ves1a variant) rather than expression of certain vas1a variants needs to be further discussed. The expression levels of the variants shown in Figure 2A reveal that the two ‘high binders’ have the higher levels of expression than the two ‘low binders’. Concern regarding levels of surface ves1a protein expression being directly related to binding affinity is also raised in the transfection experiments when the transfection of ‘low binder’ ves1a into a low binding construct also results in increased binding (although not significant). To combat this criticism, RT-PCR determined expression levels of the ves1a constructs in Fig 3C could be measured. This could determine whether the alternate binding affinities are indeed due to alternate construct sequences or merely to overexpression of any vas1a construct. An interesting control would also be to transfect the C2 construct into itself – under the ef-1a promoter. This would dissect out the roles of the sequence and the expression level of a particular ves1a sequence.

3) The mapping of the LAT does not seem to be consistent. The text states that it is on Ch2 on two of the clones, in multiple sites for one and undetermined in the fourth. However, this text does not agree with the designation of LATs (by blue and red triangles) in Figure 4B, where there are LATs discovered in each of the clones. A discussion as to how this divergence could have happened so quickly (21 rounds of selection) is warranted. As well, an overall discussion as to why the location of LATs is of scientific interest would be helpful.

Reviewer #3: Please take note of the following concerns regarding the manuscript: This manuscript is based on cloning Babesia parasites to test whether VESA1a is responsible for cytoadhesion. The description of the experiment lacks details important for the conclusion.

1) Cloning a single infected RBC from bovine brain endothelial cells containing multiple attached RBCs is not clearly explained.

2) The authors did not specify the number of passages of the Babesia culture after cloning the parasites.

3) It is unclear why the authors did not select a clone that lacked cytoadhesion ability. This would have allowed them to demonstrate the restoration of cytoadhesion capability by transfected clones.

4) The authors did not provide information on how they restored the RBC culture with the cytoadhesion clones, nor did they specify the parasitemia used for RNA and DNA isolation.

**Part III – Minor Issues: Editorial and Data Presentation Modifications**

Reviewer #1: Minor points

1. line 103. Many ves gene clusters are internal, not subtelomeric, distinguishing this parasite from Plasmodium, Trypanosoma, and many others.

2. line 136. Perhaps the authors could comment on why the Texas strain required 16 rounds of panning to detect cytoadherent parasites. As I recall, in the original description of the method (O’Connor et al. 1999. Infect. Immun.) those authors were able to recover cytoadherent parasites with only a few selections. Is the difference in the parasites or methodology? Please comment.

3. line 145. Because of their prior association with expression of the cytoadherence phenotype the authors should include information on transcript levels of the genes encoding SBP2t11 and VEAP proteins (Gallego-Lopez et al. 2018. Intl. J. Parasitol.; Hakimi et al. 2020. PLoS Pathogens), and perhaps comment on their apparent expression stability.

4. Figure 3A. Please identify the “2A” element and the locations of (relevant) stop codons. In Figure 3C please add a significance bar comparing C2 and C5.

5. line 206. As worded, it sounds as though ves1a and ves1b are mutually exclusive in their expression. Please rephrase to make it clear that different ves1a are mutually exclusive from other ves1a, and ves1b from other ves1b.

6. line 319. It is confusing that the ves1b sequences were conserved, yet variant ves1a transcripts were arising from different expression sites. Are the bidirectional “LAT”s transcribed in only one direction?

7. line 354. Please indicate whether the cDNAs cloned for use in transgenesis were sequenced to confirm their representation of the population.

8. line 700 (Figure S1). Please move the black bars indicating the CKRD to beneath the sequences so that they are underlining as stated. Currently they are above, indicating incorrect sequences.

9. Figure 1B. It might be helpful to describe for the reader what they are seeing. Without a frame of reference such an image can be confusing.

10. Figures 4A and 4B. Please orient the chromosomal sequences the same way in the dot plots and synteny plots (from Chr 1 to 4 or Chr 4 to 1 left to right in each, not both orientations). This will aid the reader in interpretation.

11. Figure 5. Please include an in-figure index of the colors to identify what each pie wedge represents.

12. Figure 6. Please indicate whether the Hoechst and GFP images were obtained as single exposures at the same focal plane or are compiled z-stacks. This affects interpretation of the location of the VESA1a-GFP polypeptides within the parasite and erythrocyte, and thus its availability to mediate cytoadherence. It would be helpful if the images could be provided at the same magnification.

Reviewer #2: Other more minor areas of concern:

Line 53 – I am not sure that the data speak at all to cytoadherence specificity. Please elaborate. I see no data discussing potential receptor ligands. That is what I think of when I think of specificity. Is something else meant by this phrase here?

Line 291 – The discussion of endothelial receptors for PfEMP1 interactions is inadequate. At a minimum EPCR binders should be mentioned.

Line 293 – The authors state that: ‘using a static cytoadhesion assay, we were not able to find novel ligands for binding of iRBCs ‘ , yet is does not seem that this was one of the aims of the paper. Did they attempt to look for other ligands and fail? If so, these data are not shown in the current manuscript.

Figure 5 – clone 5: There is no beta chain labeled in the pie chart and two alphas. Is that correct? Or are one of those supposed to be a beta?

Reviewer #3: Few typos throughout the manuscript

PLOS authors have the option to publish the peer review history of their article (what does this mean? ). If published, this will include your full peer review and any attached files.

**Do you want your identity to be public for this peer review?** For information about this choice, including consent withdrawal, please see our Privacy Policy .

Reviewer #1: No

Reviewer #2: No

Reviewer #3: No
---

## [Decision Letter · Decision Letter 1]

15 Apr 2025

Dear Dr Hakimi,

We are pleased to inform you that your manuscript '*ves1α* genes expression is the major determinant of *Babesia bovis* -infected erythrocytes cytoadhesion to endothelial cells' has been provisionally accepted for publication in PLOS Pathogens.

Best regards,

Tracey J. Lamb

Section Editor

PLOS Pathogens

Tracey Lamb

Section Editor

PLOS Pathogens

Sumita Bhaduri-McIntosh

Editor-in-Chief

PLOS Pathogens

orcid.org/0000-0003-2946-9497

Michael Malim

Editor-in-Chief

PLOS Pathogens

orcid.org/0000-0002-7699-2064

Reviewer Comments (if any, and for reference):

Reviewer's Responses to Questions

**Part I - Summary**

Reviewer #2: The authors have addressed some of the concerns raised during the first round of review. However, my main concern – that levels of ves1a rather than the specific sequence of ves1a protein determine binding levels remains. The authors have transfected ves1b transcripts into the same low binding clone and seen no increase in binding (Figure 4). This eliminates the possibility of ves1b contribution. The question of sequence vs abundance of ves1a remains. This question arises as expression is under a promoter that produces high levels of transcript throughout the life cycle. The authors have not addressed this further. The suggestion of transfecting the low-level binder C2 clone with its own ves1a gene under the ef-1a promoter was not attempted. This would be the definitive experiment to determine the role of level vs sequence. It is also unfortunate that the ideal clone – a non-binder is not available for transfection experiments. The current data does point to ves1a being responsible for binding, but with the caveat that the protein expression is non-physiologic in both timing and level.

Reviewer #3: I am satisfied with the revision. The author did a good job answering all my concerns.

Reviewer #4: The revised manuscript by Hakimi and colleagues describes the role of the VESA1 proteins in cytoadhesion of red blood cells infected by the apicomplexan parasite Babesia bovis. These parasites are a significant pathogen of cattle and are known to cause disease through circulatory disruption due to adhesion of infected cells to the vascular endothelium. This is analogous to the pathology caused by the human malaria parasite Plasmodium falciparum. It has been recognized for some time that cytoadhesion is caused by display of the VESA1 heterodimeric protein on the infected cell surface, but additional details regarding these interactions have not been described. Here the authors employ molecular genetic manipulation of cultured parasites to determine that it is the VESA1a protein that mediates selective adhesion to brain endothelial cells while the VESA1b protein appears to likely provide support for this interaction. This is a significant insight into the pathology of this important pathogen.

The study is quite simple and yet convincing in its conclusions. There are very few groups working on this pathogen compared to other apicomplexan organisms, thus the finding marks an important advance. The previous version of the paper received mostly positive reviews, although several details required greater explanation or clearer description to justify the authors’ conclusions. The authors have suitably addressed these concerns. The one remaining caveat to their conclusions is the possibility that changes in the cytoadhesive properties of the infected cells is not due to changes in the VESA1a gene that is expressed (which is the authors’ favored conclusion), but rather in the expression level of VESA1a. There is evidence for this in several figures, including Figure 3C. This alternative conclusion was pointed out by two of the previous reviewers. The authors have conceded this point by addressing it in several places in the text and suggesting that a better experimental system will be required to directly address this possibility. This is acceptable. In addition, this alternative interpretation does not detract from the overall conclusion that VESA1a is the subunit responsible for cytoadhesion in this system.

**Part II – Major Issues: Key Experiments Required for Acceptance**

Reviewer #2: See above

Reviewer #3: None

Reviewer #4: None.

**Part III – Minor Issues: Editorial and Data Presentation Modifications**

Reviewer #2: See above

Reviewer #3: Italicize the scientific name of the in the references.

Reviewer #4: 1. In Figure 5, the authors show both primary and secondary LATs for the C2 clone in panel C, but only show a primary LAT for this clone in panel B. Is this intentional?

2. In the Discussion section on lines 367-369, the authors state “The frequent recombination characteristic of this multigene family is likely limited to ves1a which is explained by the function of VESA1a for cytoadhesion.” I don’t understand the logic here. Variation through recombination is generally thought of as a way to avoid antibodies (antigenic variation), which is not necessarily related to cytoadhesion. Thus, it is likely that VESA1b, if it is antigenic, will undergo recombination regardless of whether it contributes directly to cytoadhesion. The fact that the parasite maintains a large repertoire of vesa1b genes within its genome is consistent with this idea.

PLOS authors have the option to publish the peer review history of their article (what does this mean? ). If published, this will include your full peer review and any attached files.

**Do you want your identity to be public for this peer review?** For information about this choice, including consent withdrawal, please see our Privacy Policy .

Reviewer #2: No

Reviewer #3: No

Reviewer #4: No

---

## [Editor Report · Acceptance letter]

Dear Dr Hakimi,

We are delighted to inform you that your manuscript, "*ves1α* genes expression is the major determinant of *Babesia bovis* -infected erythrocytes cytoadhesion to endothelial cells," has been formally accepted for publication in PLOS Pathogens.

Best regards,

Sumita Bhaduri-McIntosh

Editor-in-Chief

PLOS Pathogens

orcid.org/0000-0003-2946-9497

Michael Malim

Editor-in-Chief

PLOS Pathogens

orcid.org/0000-0002-7699-2064